# Effects of 8-Week Electromyostimulation Training on Upper-Limb Muscle Activity and Respiratory Gas Analysis in Athletes with Disabilities

**DOI:** 10.3390/ijerph20010299

**Published:** 2022-12-24

**Authors:** Jongbin Kim, Joonsung Park, Jeongok Yang, Youngsoo Kim, Inhyung Kim, Himchan Shim, Changho Jang, Mincheol Kim, Myeongcheol Kim, Bomjin Lee

**Affiliations:** 1Division of Kinesiology, Silla University, Busan 46958, Republic of Korea; 2Independent Researcher, Busan 48316, Republic of Korea; 3Independent Researcher, Busan 48499, Republic of Korea

**Keywords:** disabled athletes, disability, electromyostimulation, respiratory gas analysis

## Abstract

This study was aimed at verifying the efficacy of EMS training by investigating the changes in upper-limb muscle functions and energy expenditure in athletes with disabilities after an 8-week intervention of EMS training. We compared variations in muscle activity, respiratory gas, and symmetry index (SI) after an 8-week intervention in eight professional male athletes with disabilities wearing an electromyostimulation (EMS) suit (age: 42.00 ± 8.67 years, height: 1.65 ± 0.16 m, weight: 64.00 ± 8.72 kg, career length: 11.75 ± 3.83 years). For EMS training, each participant wore an EMS suit. EMS was applied to the upper-limb muscles pectoralis major and triceps at 40 °C water temperature, with a 25 Hz frequency (duty cycle 10%) for 15 min, followed by a 5 Hz frequency (duty cycle 5%) for 5 min. The pre- and post-intervention measurements were taken in the same way at a pre-set time (for 1 h, twice a week) for 8 weeks. Training involved a seated chest press, and the muscle activity (pectoralis major, triceps, and antebrachial muscles), upper-limb SI, and respiratory gas variables (maximal oxygen consumption (VO_2_), carbon dioxide output (VCO_2_), respiratory quotient (RQ), metabolic equivalents (METs), and energy expenditure per min (Energy expended per minute; EEm)) were analyzed. Variations pre- and post-intervention across the measured variables were analyzed. Regarding the change in muscle activity, significant variations were found in the pectoralis major right (*p* < 0.004), pectoralis major left (*p* < 0.001), triceps right (*p* < 0.002), and antebrachial right (*p* < 0.001). Regarding left-to-right SI, a positive change was detected in the pectoralis major and triceps muscles. Additionally, respiratory gas analysis indicated significant variations in VO_2_ (*p* < 0.001), VCO_2_ (*p* < 0.001), METs (*p* < 0.001), and EEm (*p* < 0.001). EMS training improved muscle strength and respiratory gas variables and is predicted to contribute to enhanced muscle function and rehabilitation training for athletes with disabilities.

## 1. Introduction

Exercise can enhance the quality of life through improved mental health and physical fitness [1,2,3] and increased life expectancy [4]. Exercise can also prevent chronic diseases such as diabetes, osteoporosis, and cardiovascular disease by promoting muscle development [5] and calorie consumption [6]. Hence, the importance of exercise is recognized by both individuals with and without disabilities. However, for individuals with disabilities, the innate or acquired damage to motor and sensory nerves prevents exercise due to the discomfort related to reduced mobility [7]. If this could be overcome so that individuals with disabilities could perform regular exercise, the exercise would serve as a critical means not only of improving health and maintaining physical fitness but also of rehabilitation and functional recovery [8].

Recently, athletes with disabilities have further improved in performance through the Paralympics and other opportunities; therefore, the difference from professional athletes without disabilities is no longer prominent, while the emphasis is now placed on the highest athletic performance [9,10]. Nevertheless, in the case of athletes with disabilities, their physical strength and ability are lower than those of non-disabled athletes due to limitations in environmental factors such as training grounds and convenience facilities for participation in sports, constraints in economic factors, and the lack of time, affecting training and performance [11]. This indicates the need to ensure that athletes with disabilities maintain certain performance levels regarding aerobic and anaerobic muscular abilities [12]. As a result, there is an urgent need for suitable venues and assistive devices to help with the improvement of muscle strength, energy expenditure, and the respiratory quotient (RQ) in athletes with disabilities.

Among various assistive devices, the electromyostimulation (EMS) device has been clinically accepted for rehabilitation purposes [13]. The EMS device has recently been recognized as a method to induce changes in the active potential based on the involuntary contraction of the skeletal muscles classified as voluntary muscles, and the use of EMS training in muscles has been reported to effectively increase muscle strength and the joint range of motion [14]. In addition, repeated EMS on muscles has been reported to increase capillary vessels and the flexibility of blood vessels in muscle fibers, as well as the blood flow [15], enhance the muscle strength [16], increase the motor units at nerve root junctures [17,18], and restore the damaged functions of the central nervous system [19,20]. As such, low-frequency EMS has attracted much attention as a new method of exercise [21], and EMS training in athletes has been shown to effectively enhance muscle strength [22,23,24]. However, in contrast to the positive findings on the effects of EMS in muscle development, the reported drawbacks include fatigue of the neuromuscular system due to excessive involuntary muscle contractions caused by electrical stimulation and sudden muscle fatigue caused by conflicting orders of muscle recruitment [14,25]. 

While systematic training for athletes with disabilities should reflect the scientific principles and medical outcomes and concerns to a greater degree compared to athletes without disabilities [26,27], it is also truly important to address the limitations felt by athletes with disabilities regarding the types and levels of disabilities, the lack of the systematization of training, and psychological stress [28]. The range of possible upper-limb motor functions varies substantially according to the affected area in athletes with disabilities, and continuous stimulation should be applied to the muscles due to the limited range of possible motions caused by degenerated muscles and nerves [29]. Among upper-limb joint injuries in athletes with disabilities, injury of the shoulder joint that constitutes the rotator cuff based on the scapular was reported to cause markedly reduced upper-limb muscle strength and range of motion (ROM) [30,31]. EMS training on such injuries has been shown to improve the involuntary upper-limb motions [32]. Furthermore, Alon and Levitt [33,34] reported that EMS training on the wrist and finger muscles in stroke patients could improve upper-limb function. In addition, EMS training was reported to have positive effects on the physical composition and muscle function in menopausal women (Kemmler et al.; 2010); when the energy consumption was compared according to the use of EMS in identical participants, the level was approximately 20% higher in the EMS training group, indicating a high level of exercise efficiency [23]. However, participants in previous studies regarding whole-body-EMS were older adults [35,36]; therefore, a greater variety of participants should be investigated. It was suggested that a novel exercise program could be developed using EMS training to replace conventional exercise programs, especially for individuals who do not have time to exercise or dislike exercise itself, as well as those who are faced with a challenge in performing exercises due to conditions such as sarcopenia and osteoporosis [23,24].

Most previous studies were clinical studies conducted on athletes without disabilities or older adults regarding the validation of EMS training effects such as enhanced muscle strength, reduced pain at the site of injury, and increased fatigue of the neuromuscular system. There is a general lack of studies on the effects of EMS training in enhancing muscle function. Thus, this study investigated pre–post changes in upper-extremity muscles (pectoralis major, triceps, and antebrachial), the bilateral symmetry index, and gas respiration (VO_2_) before and after 8 weeks of EMS training to verify the effects of EMS training in athletes with disabilities. VCO_2_, RQ, METs, and EEm were analyzed.

## 2. Materials and Methods

### 2.1. Participants

We recruited volunteers after explaining the purpose of the study and the measurement procedure to athletes (swimming, weightlifting, and running) participating in the National Paralympic Games. Eight athletes (six with paraplegia and two with physical disabilities) with normal scores were selected. Table 1 shows the physical characteristics of all participants. Measurements were obtained after Bioethics Committee approval (approval no.: 1041449-202206-HR-002).

### 2.2. Procedures

The athletes with disabilities participating in this study were given an explanation on the study procedures and the goals of the measurements, after which they signed and submitted written consent. To determine the effects of an 8-week intervention of EMS training, the pre- and post-intervention variables were measured with the seated chest press (Miniplus, Ronfic, Korea). First, the participants performed a light warm-up exercise for 5 min, and upper-limb muscle stiffness was measured in the sitting position. The participants were guided to sit before the seated chest press device with their hands holding the bar, their forearms parallel to the floor, their chin pulled toward their chest, and their line of sight adjusted to 15° forward. The participants then lightly grasped the grip with their elbows directed outward, and, while breathing out in a relaxed manner, they pushed their elbow joints outward through extension. This was followed by slow flexion as the participants breathed in. The cable direction was set to allow for the pull at an identical 90° position from the ground by all participants. To examine the changes in muscle activity, six surface electrodes were attached to the left and right pectoralis major, triceps, and antebrachial muscles, and the maximum voluntary isometric contraction (MVIC) was measured at 60% of the participant’s weight for 5 s. Next, preparations for muscle activity and respiratory gas measurements were made by applying a soft nylon mask on the nose and mouth of the participant and maximally tightening the head harness to prevent the escape of the inspiratory and expiratory gases. In addition, three sets of seated chest presses were performed by the participants five times, in the same posture as previously described. The rest between each set was 3 min. For EMS training, each participant wore an EMS suit, and at a 40 °C water temperature, the frequency was set to 25 Hz (duty cycle 10%) for 15 min and then 5 Hz (duty cycle 5%) for 5 min, with the EMS applied to the upper-limb muscles (pectoralis major and triceps). The pre- and post-intervention measurements were taken in the same way at a pre-set time (for 1 h, twice a week) for 8 weeks.

### 2.3. Data Processing and Analysis

#### 2.3.1. Muscle Stiffness Analysis

Prior to the measurements, participants were given time to take an adequate rest. To measure the stiffness of the muscles during the upper-limb motions in athletes with disabilities, the participants were guided to sit on a chair in a relaxed manner to perform the MMT. Then, in a manner that does not cause pain or muscle fatigue, and with adequate rest and explanations of the directions, the participants were guided to perform shoulder joint flexion and extension, shoulder joint abduction and adduction, and heel joint flexion and extension. Each motion was maintained for 10 s and repeated three times to measure the stiffness at the upper-limb joints. Figure 1 describes each motion. The rating of the measurements was as follows: Normal, if the participant could overcome strong resistance in a sitting position; Good, if the resistance was weak; Fair, for the motions without resistance in a sitting position; Poor, for the motions without resistance in a hanging to prone position; Trace, if contraction was detected upon examining the antagonistic muscles. All participants displayed a level of Good or above.

#### 2.3.2. Muscle Activity Analysis

To measure muscle activity, electromyography (EMG) (Noraxon Inc. U.S.A) was used. At three muscle sites on both arms (pectoralis major, triceps, and antebrachial muscles) (Figure 2), the hair was removed, the skin surface was wiped with alcohol and then calmed, and six surface electrodes were attached. The distance between two electrodes was 2 cm, while the position was set in reference to the EMG manufacturer’s guidelines (SENIAM Guideline). The EMG signals were collected at 2000 Hz/s, and the raw data were filtered at a 40–450 Hz bandpass and then rectified. For smoothing, the root mean square (RMS) was used, and the time window was set as 50–100 ms. With the EMG connected to the surface electrodes, the MVIC was measured with actual motions of the seated chest press at 60% of the participant’s body weight. The data of 5 s were applied with the elbow joint extension. The raw data of the seated chest press performed in three sets five times were collected for each muscle type, and the mean values were used in the analysis. The equation for standardization was as follows:(1)$$Muscle\;activation=\frac{EMG_{raw}}{EMG_{MVIC}}\times 100\left(\%\right)$$

*EMG_raw_*: RMS of muscle activity upon movement

*EMG_MVIC_*: RMS of muscle activity upon MVIC.

#### 2.3.3. Left-to-Right Symmetry Index

To determine the left-to-right symmetry index (*SI*), the variables of all muscles on the left and right arms were computed as follows:(2)$$SI=\frac{\left|X_{R}-X_{L}\right|}{\frac{1}{2}\left(X_{R}+X_{L}\right)}\times 100\%$$

*X_R_*: muscles on the right, *X_L_*: muscles on the left

Here, the *SI* values ranged from the lowest (0%) to the highest (200%), with values closer to 0% indicating a higher level of symmetry [37].

#### 2.3.4. Respiratory Gas Analysis 

In the respiratory gas analysis (K5, COSMED, Italy) (Figure 2), maximal oxygen consumption (VO_2_), carbon dioxide output (VCO_2_), respiratory quotient (RQ), metabolic equivalents (METs), and energy expenditure per min (Energy expended per minute; EEm) were selected as the main variables for the use of a wireless respiratory gas analyzer, which represent the metabolic energy expenditure. The mean values of the respiratory gas data obtained during the five trials of the three seated chest presses were used. Steady state was defined as a state showing a 100 mL/min difference between the mean of the last 1 min of oxygen consumption per min and the mean of the preceding 1 min [37]. To adequately remove the noise but retain the dynamic changes in the data of all metabolic variables, the moving average filtering at a 15-window size was applied, as it was verified to have high validity by Robergs et al. [38].

### 2.4. Statistical Analysis

Statistical analyses were performed using SPSS 20.0 for descriptive statistics. The descriptive statistics and homogeneity test were conducted to analyze the general characteristics of the participants. Paired-samples *t*-tests were performed to analyze the changes in muscle activity and respiratory gas parameters, and effect sizes (Cohen’s d) were calculated. The significance level for all data was an α of 0.05.

## 3. Results

In this study, an 8-week intervention of EMS training was performed on athletes with disabilities. The consequent changes in upper-limb muscle functions and respiratory gas variables were analyzed to verify the effects of the EMS training.

### 3.1. Muscle Activity Analysis

Table 2 presents the changes in muscle activity on the left and right arms (pectoralis major, triceps, and antebrachial muscles) of athletes with disabilities upon seated chest presses, before and after the EMS training for 8 weeks. The pre- and post-intervention variations were statistically significant in the pectoralis major right (RT) (t = −4.844, *p* < 0.004), pectoralis major left (LT) (t = −0.4.073, *p* < 0.001), triceps RT (t = −4.277, *p* < 0.002), and antebrachial RT (t = −4.046, *p* < 0.001), while no significant variation was found in the triceps LT (t = −1.451, *p* < 0.100) (Figure 3).

### 3.2. Left-to-Right SI

Table 3 presents the changes in the left-to-right SI (pectoralis major, triceps, and antebrachial muscles) of athletes with disabilities upon seated chest presses, before and after EMS training for 8 weeks. The pre- and post-intervention variations were not statistically significant in the pectoralis major (t = −3.697, *p* < 0.084), triceps (t = −1.723, *p* < 0.167), and antebrachial (t = 3.997, *p* < 0.078) muscles (Figure 3).

### 3.3. Respiratory Gas Analysis

Table 4 presents the changes in the respiratory gas variables (VO_2_, VCO_2_, RQ, METs, and EEm) of athletes with disabilities upon seated chest presses, before and after EMS training for 8 weeks. The pre- and post-intervention variations were statistically significant for the VO_2_ (t = −3.293, *p* < 0.001), VCO_2_ (t = −3.245, *p* < 0.002), METs (t = −3.618, *p* < 0.001), and EEm (t = −3.318, *p* < 0.001), whereas no significant variation was found for the RQ (t = −1.291, *p* < 0.104).

## 4. Discussion

This study was conducted to investigate the changes in upper-limb muscle functions and energy expenditure in athletes with disabilities through an 8-week intervention of EMS training, with the aim being to verify the efficacy of EMS training.

EMS training was developed as an assistive device for short-term whole-body training. While the participants in previous studies were older adults [23,34,35], the participants in this study were athletes with disabilities. EMS training given to athletes with disabilities was shown to induce significant variations in the activity of the pectoralis major RT (*p* < 0.004), pectoralis major LT (*p* < 0.001), triceps RT (*p* < 0.002), and antebrachial RT (*p* < 0.001) muscles when the pre- and post-intervention muscle activity with seated chest presses were examined. The triceps LT and antebrachial LT muscles showed an increase in activity by 17% and 18%, respectively, despite the lack of significant variations, which indicated a positive effect. Previous studies on the changes in muscle strength upon EMS training included studies reporting enhanced muscle strength in athletes [38,39], those reporting an increase in muscle strength for the lower-limb and trunk maximum extension in female older adults, and those reporting an increase in muscle mass and strength in female older adults at a risk of sarcopenia following a 54-week intervention with whole-body-EMS training [23,35]. In addition, EMS (30 min a day) applied to patients with a syndromic disease was clinically verified to decrease the disease severity, which enhanced the quality of life and reduced the risk of onset [40]. In another study investigating the effects of trunk stabilization exercise and low-frequency EMS in patients with chronic back pain, the index of functional disability for back pain was shown to decrease [41]. Gradually, regular low-frequency EMS training is indicated to enhance the isokinetic muscle function and promote the development of upper-limb muscles, with an assistive role in enhancing muscle strength in athletes with disabilities, for whom fluent motions are challenging. EMS training is also predicted to have positive effects on lower-limb muscles in addition to upper-limb muscles, as it can increase exercise ability by continuously inducing muscle contractions.

While left-to-right muscle asymmetry results from an injury to the musculoskeletal system [42,43], the left-to-right SI for upper-limb muscles measured in this study did not show significant variations in the pectoralis major, triceps, and antebrachial muscles. However, in the pectoralis major and triceps muscles, EMS training was effective, as the left-to-right SI increased by 36% and 39%, respectively. Following low-frequency EMS training for 8 weeks, the pectoralis major and triceps muscles were confirmed to have improved left-to-right symmetry. A comparison of the bilateral motor loss between 40 right-dominant and 40 left-dominant people found that muscle strength and activity differed significantly [44]. Although motor loss appeared to be low, it was bilateral. In addition, [45] showed that the asymmetry index decreased when using a power-assisted wheelchair; however, antebrachial muscles without low-frequency EMS displayed asymmetry. This suggested that regular low-frequency EMS could lead to the symmetry of left and right muscles, with predicted positive effects in preventing injuries and reducing pain. 

Respiratory gas analysis with the seated chestpress after the 8-week EMS training in athletes with disabilities showed significant variations in VO_2_ (*p* < 0.001), VCO_2_ (*p* < 0.001), METs (*p* < 0.001), and EEm (*p* < 0.001). VO_2_ and VCO_2_ are variables of cardiopulmonary fitness [46]; in a study conducted on sedentary patients, EMS training was shown to increase the VO_2_ by approximately 20% [45]. The increase in VO_2_ in this study was 43%, indicating an increase of >4-fold, which suggested an improvement in cardiopulmonary fitness. Similar to the results of this study, VO_2_ (42 times) and VCO_2_ (47 times) also increased, and the increase in the rate of CO_2_ production was higher than that of oxygen intake [47]. In addition, the VCO_2_ is an indicator of CO_2_ production, as the oxygen demand upon muscle contraction is satisfied during exercise because of the excitation of the inspiratory and expiratory systems [48]. The VCO_2_ in this study showed a significant variation with EMS training, which is thought to be due to the cardiopulmonary fitness being improved based on the increased VO_2_ in athletes with disabilities after EMS training. Therefore, EMS training is presumed to promote more efficient muscle contractions through an increased VCO_2_.

METs, measured by a wireless respiratory gas analyzer, are categorized based on the intensity of physical activity performed by adults as follows: low-intensity (~3 METs), moderate-intensity (3–6 METs), and high-intensity (6 METs) [49]. In this study, the 8-week EMS training led to a 1 METs increase. In the study by Lee et al. [50], where the gait intensity (METs) was measured at speeds of 3.2 km/ h, 4.8 km/h, and 5.6 km/h on the treadmill for 52 adults in their 20s (23 males and 19 females), the mean METs per speed were 3.46 ± 0.45, 4.70 ± 0.59, and 5.69 ± 0.69, respectively. This lent support to the change in METs through continuous EMS training in athletes with disabilities, and exercise intensity was presumed to be low to moderate based on the EMS training.

Lastly, the EEm in the respiratory gas analysis increased from 2.20 ± 0.82 kcal/min to 3.19 ± 1.01 kcal/min after the 8-week EMS training, which agreed with the observed increase of 1–2 kcal/min through EMS in a previous study [51]. In addition, Scott et al. [52] claimed that an increase in EEm could be achieved through EMS training, and Hamada et al. [53] mentioned that EMS was deeply associated with EEm. Therefore, regular EMS training is presumed to promote cardiopulmonary fitness and energy expenditure through increased muscle mass. Regular EMS training given to athletes with disabilities for 8 weeks had activated the calorie consumption to a high level, even in the absence of exercise; therefore, it is predicted that EMS training without exercise could solve the problem of obesity risk factors in individuals with disabilities, for whom motions are not fluent, through the generation of energy expenditure. 

## 5. Conclusions

In this study, athletes with disabilities underwent EMS training for 8 weeks, and the consequent changes in upper-limb muscle functions and energy expenditure were examined. The conclusions are as follows: First, EMS significantly increased muscle activity in the pectoralis major RT, pectoralis major LT, triceps RT, and antebrachial RT. Second, EMS induced significant variations in the VO_2_, VCO_2_, METs, and EEm. Third, no significant variation in the left-to-right SI was found for the pectoralis major, triceps, and antebrachial muscles through EMS training. EMS training in this study was shown to have a significant short-term training effect on upper-limb muscle activities for athletes with disabilities, which would lead to positive effects for muscle strength and cardiopulmonary fitness and enhance athletic performance. Athletes in a greater diversity of fields should be investigated, while the changes in training effects on the upper- and lower-limb muscles and the changes in the effects of aerobic exercise should be determined, for which future follow-up studies will be conducted to continue the academic research on EMS training. In the future, it is suggested to study the changes in the muscles of the upper and lower extremities.

## Figures and Tables

**Figure 1 ijerph-20-00299-f001:**
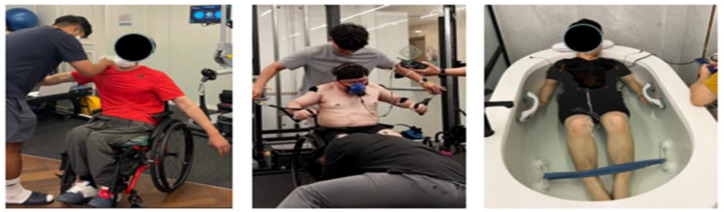
Manual muscle test (MMT) (left), electromyography (EMG) and respiratory gas measurements (middle), and EMS suit training (right).

**Figure 2 ijerph-20-00299-f002:**
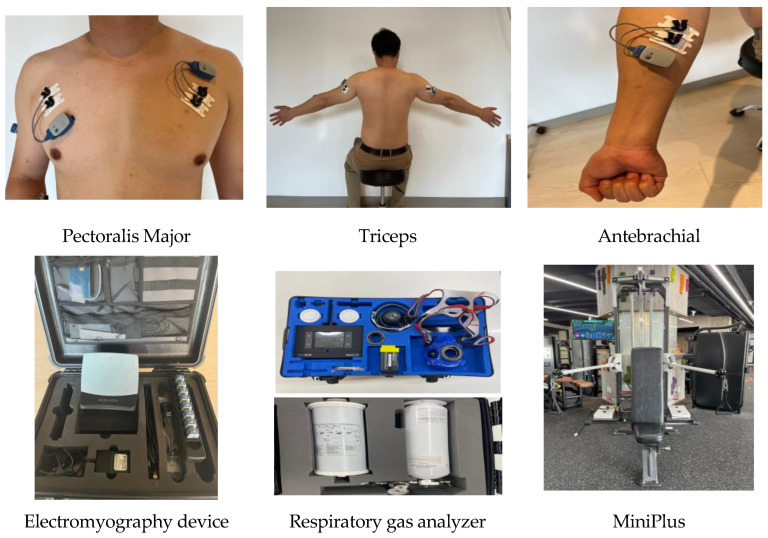
The sites of electrode attachment for the electromyography (EMG) and the devices used in this study.

**Figure 3 ijerph-20-00299-f003:**
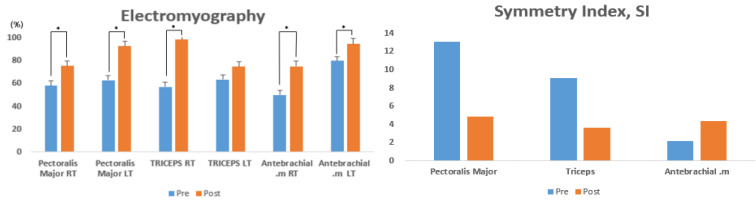
Comparison of muscle activity pre- and post-intervention with 8 weeks of EMS training.

**Table 1 ijerph-20-00299-t001:** The physical composition of the participants.

Variable	Age (years)	Height (m)	Weight (kg)	Duration of Athletic Career (years)
Mean ± SD	42.00 ± 8.67	1.65 ± 0.16	64.00 ± 8.72	11.75 ± 3.83

**Table 2 ijerph-20-00299-t002:** Changes in muscle activity pre- and post-intervention on the left and right arms.

Variables	Pre	Post	t	Effect Size	*p*
	Mean ± SD	Mean ± SD			
Pectoralis Major RT	58.22 ± 21.78	75.31 ± 18.06	−4.844	0.84 ^L^	0.004 *
Pectoralis Major LT	62.83 ± 21.87	92.64 ± 20.48	−4.073	1.33 ^L^	0.001 *
Triceps RT	57.09 ± 24.13	98.58 ± 54.42	−4.277	0.84 ^L^	0.002 *
Triceps LT	63.71 ± 17.02	74.73 ± 35.23	−1.451	0.44 ^S^	0.100
Antebrachial RT	50.27 ± 14.81	75.13 ± 27.61	−4.046	1.16 ^L^	0.001 *
Antebrachial LT	79.86 ± 41.45	94.77 ± 34.49	−1.167	0.27 ^S^	0.095

LT: left, RT: right. * *p* < 0.05, effect size: ^S^ small (~0.2), ^L^ large (~0.8).

**Table 3 ijerph-20-00299-t003:** Left-to-right symmetry index (SI, %).

Variables	Pre	Post	t	Effect Size	*p*
	Mean ± SD	Mean ± SD			
Pectoralis Major	13.13 ± 2.42	4.85 ± 3.14	0.189	0.28 ^S^	0.441
Triceps	9.12 ± 2.89	3.63 ± 2.34	0.942	0.46 ^S^	0.260
Antebrachial	2.20 ± 1.06	4.33 ± 2.51	1.001	0.88 ^L^	0.250

Effect size: ^S^ small (~0.2), ^L^ large (~0.8).

**Table 4 ijerph-20-00299-t004:** Changes in respiratory gas variables pre- and post-intervention.

Variables	Pre	Post	t	Effect Size	*p*
	Mean ± SD	Mean ± SD			
VO_2_ (mL/min)	448.79 ± 163.46	645.69 ± 208.87	−3.293	1.10 ^L^	0.001 *
VCO_2_ (mL/min)	407.28 ± 172.92	611.08 ± 198.74	−3.245	1.14 ^L^	0.002 *
RQ	0.92 ± 0.11	0.98 ± 0.17	−1.291	0.36 ^S^	0.104
METs	2.09 ± 0.68	3.05 ± 1.04	−3.618	1.15 ^L^	0.001 *
EEm (kcal/min)	2.20 ± 0.82	3.19 ± 1.01	−3.318	1.13 ^L^	0.001 *

VO_2_: peak oxygen uptake, VCO_2_: carbon dioxide production, RQ: respiratory quotient, METs: metabolic equivalents, EEm: energy expenditure per minute. * *p* < 0.05, effect size: ^S^ small (~0.2), ^L^ large (~0.8).

## Data Availability

Not applicable.

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
