# Peer review of "Effects of 8-Week Electromyostimulation Training on Upper-Limb Muscle Activity and Respiratory Gas Analysis in Athletes with Disabilities"

_ijerph, 2022, doi:10.3390/ijerph20010299_

Round 1
Reviewer 1 Report
A quasi-experimental study with pre- and post-intervention measurements of an 8-week electrostimulation training programme in athletes with disabilities, evaluating the effects on upper body muscle activity and respiratory gases.
Although I find this study appropriate and interesting, there are some considerations that need to be addressed in order to be accepted.
Comments:
Title and abstract
The design followed by the authors has not been included, neither in the title nor in the abstract. I would recommend the authors to include, in the title or abstract, the type of design used in this study.
The participants are male athletes with disabilities, but there is no contextualisation of the type of sport practised, nor the level of the athletes. If they compete in some kind of high level Paralympic sport, or recreationally, ... I would recommend providing more information.
In the summary, the type of stimulus does not appear, nor the intensity of the impulse applied with EMS. Nor is there any reference to the exercises performed during the training programme.
Introduction
I believe that they have done a good job of substantiating the rationale for this study. Although the authors cite the objectives of the study, I feel that a study hypothesis to verify or reject should have been stated.
Method
Participants
I have the same doubts as in the abstract. It would be necessary to report on the level of the athletes and the type of sport practised, among others.
There is no reference to recruitment.
How was the sample recruited, was any kind of sampling method applied to recruit the sample, what were the eligibility criteria? Inclusion and exclusion criteria are not clear, was it a pilot study, and was there no possibility of having a control group?
Statistical analysis
I have some doubts about the statistical power and effect size. Could it be calculated?
Results
Could the effect size be included in the tables?
Discussion
I suggest extending the discussion, including the limitations of the study, as well as expanding a little more on practical applications.
Conclusions
These conclusions are in line with the findings and the methodology used.
Author Response
감사합니다 리뷰어의 세심한 리뷰는 완벽한 논문으로 이어졌습니다. 다시 한 번 감사드립니다.

Reviewer 2 Report
Overall, I found the manuscript organized and presented well. The topic is interesting and navel.
Please note that the comments below are meant to assist in the development of your manuscript.
Introduction:
L 34: on motor or to motor?
L 35 – 38: check punctuation of the sentence structure
L 39: sportsmanship should be changed. Sportsmanship is the act of being a good sport. I believe a different word choice is needed.
L42-46: Please consider breaking this long sentence into at least two if not three.
L63-65: I am not sure this sentence is needed here or it could be placed earlier in the paragraph
General comment: There are several long and drawn out sentences. Some spanning five page lines. Please consider breaking these up to allow the consumer to ingest the information in smaller quantities.
L90: Remove ‘Most’. Previous clinical studies….
Materials and Methods:
L100: Is B Metropolitan City an actual name
What is the relevance of a ‘Career Length’?
Were there differing types of injuries? Grades? Locations?
What system, specifically, was used to assess gases?
Overall, there is a need to develop a greater level of transparency through additional information.
Results:
No issues
Discussion:
There seems to be a consistent explanation of the results and not a terribly in depth discussion of how the results relate to previous research. Some research is cited, but more would benefit this section.
The statement regarding future research should be moved to the Conclusions section.
Conclusions:
No issues
References:
Formats are slightly inconsistent.
Author Response
thank you The reviewer's meticulous review resulted in a perfect thesis. Thank you again.

Round 2
Reviewer 1 Report
I would recommend authors to respond to reviewers' comments in English and not in Korean. Nevertheless, they have done a good job and I have no objection to this manuscript being accepted.